# Atomic resolution cryo-EM structure of a native-like CENP-A nucleosome aided by an antibody fragment

Bing-Rui Zhou [1,5], K. N. Sathish Yadav[2,5], Mario Borgnia[3], Jingjun Hong[1], Baohua Cao[2], Ada L. Olins[4], Donald E. Olins[4], Yawen Bai[1] & Ping Zhang[2]

Genomic DNA in eukaryotes is organized into chromatin through association with core histones to form nucleosomes, each distinguished by their DNA sequences and histone variants. Here, we used a single-chain antibody fragment (scFv) derived from the anti-nucleosome antibody mAb PL2-6 to stabilize human CENP-A nucleosome containing a native α-satellite DNA and solved its structure by the cryo-electron microscopy (cryo-EM) to 2.6 Å resolution. In comparison, the corresponding cryo-EM structure of the free CENP-A nucleosome could only reach 3.4 Å resolution. We find that scFv binds to a conserved acidic patch on the histone H2A-H2B dimer without perturbing the nucleosome structure. Our results provide an atomic resolution cryo-EM structure of a nucleosome and insight into the structure and function of the CENP-A nucleosome. The scFv approach is applicable to the structural determination of other native-like nucleosomes with distinct DNA sequences.

[1] Laboratory of Biochemistry and Molecular Biology, Center for Cancer Research, National Cancer Institute, National Institutes of Health, Bethesda, MD 20892, USA. [2] Structural Biophysics Laboratory, Center for Cancer Research, National Cancer Institute, National Institutes of Health, Frederick, MD 21702, USA. [3] Genome Integrity and Structural Biology Laboratory, National Institute of Environmental Health Sciences, National Institutes of Health, Research Triangle Park, NC 27709, USA. [4] Department of Pharmaceutical Sciences, College of Pharmacy, University of New England, Portland, ME 04103, USA. [5] These authors contributed equally: Bing-Rui Zhou, K. N. Sathish Yadav. Correspondence and requests for materials should be addressed to Y.B. (email: BaiYaw@mail.nih.gov) or to P.Z. (email: ping.zhang@nih.gov)

Nucleosomes are substrates of the protein machineries responsible for DNA replication, recombination, transcription, repair, and chromosome segregation. Nucleosome core particles (NCP) comprise an octamer of two copies of each of the four histones (H2A, H2B, H3, and H4) wrapped with ~145–147 bp DNA[1–3]. The ability to obtain high-quality diffracting crystals for X-ray crystallography analysis is strongly dependent on the DNA fragment used for NCP assembly[2]. Single particle cryo-EM provides an alternative way to determine atomic resolution structures without requiring crystals; however, nucleosomes tend to dissociate during cryogenic sample preparation, which apparently limits the resolution of structures of NCPs determined[4]. To date, single particle cryo-EM studies of free nucleosomes or those in complex with other proteins have mainly relied on the Widom "601" (W601) DNA, which was selected in vitro for high affinity binding to the core histones[5].

Here, we overcome this hurdle by using a single-chain antibody fragment (scFv) to stabilize the nucleosome. We determine the cryo-EM structure of the human centromeric nucleosome containing CENP-A and a native α-satellite DNA sequence at 2.6 Å resolution. Our study reveals structural features and provides insights into the structure and function of the centromeric nucleosome. The results and the scFv method present here pave an avenue for the structural determination of nucleosomes with natural DNA sequences at atomic resolution by cryo-EM.

## Results

**An antibody fragment and nucleosome stabilization.** Antibodies and their fragments have often been used to stabilize macromolecules for structural determination[6]. Anti-nucleosome antibodies are present in patients with systemic lupus erythematosus autoimmune disease. Notably, the PL2-6 antibody, isolated from autoimmune mice with lupus-like nephritis[7], has been suggested to bind the conserved acidic patch of the nucleosome surface[8]. We engineered a single-chain fragment (scFv) from the PL2-6 antibody, which includes the variable heavy (Hv) and light (Lv) chains connected by a flexible linker[9] (Fig. 1a and Supplementary Fig. 1a). Gel shift assay and isothermal titration calorimetric experiments showed that scFv bound to the NCP containing *Drosophila* core histones and 147 bp W601 DNA, termed $NCP^{H3, W601}$, with 2:1 stoichiometry and a dissociation constant (Kd) of ~190 nM for each binding site (Supplementary Fig. 1b, c). Indeed, the scFv-$NCP^{H3, W601}$ complex particles distributed homogenously in the vitrified ice as intact particles without observable dissociation, whereas the free $NCP^{H3, W601}$ showed substantial dissociation (Supplementary Fig. 1d). We obtained a density map of the scFv-$NCP^{H3, W601}$ complex with an overall resolution of 3.0 Å and determined its structure (Supplementary Figs. 2, 3 and Supplementary Table 1).

To investigate whether scFv can also stabilize nucleosomes with native DNA sequences, we reconstituted the human centromeric NCP containing CENP-A (a H3 variant) and a native 145 bp α-satellite (NAS) DNA[10], termed $NCP^{CENP-A, NAS}$. Indeed, scFv stabilized the NCP not only when exposed to increased salt concentration in solution (Supplementary Fig. 4c) but also during the vitrification process (Supplementary Fig. 4d). We obtained a density map at an overall resolution of 2.6 Å for the scFv-$NCP^{CENP-A, NAS}$ complex (Fig. 1, Supplementary Figs. 5, 6, Supplementary Movie 1, and Supplementary Table 1) and solved its structure (Fig. 2a, b).

**Interactions between the nucleosomes and the scFv.** In both structures of the scFv-$NCP^{CENP-A, NAS}$ and scFv-$NCP^{H3, W601}$ complexes, the residues in the variable loops of the scFv interact with the H2A–H2B region including the acidic patch (Figs. 1b

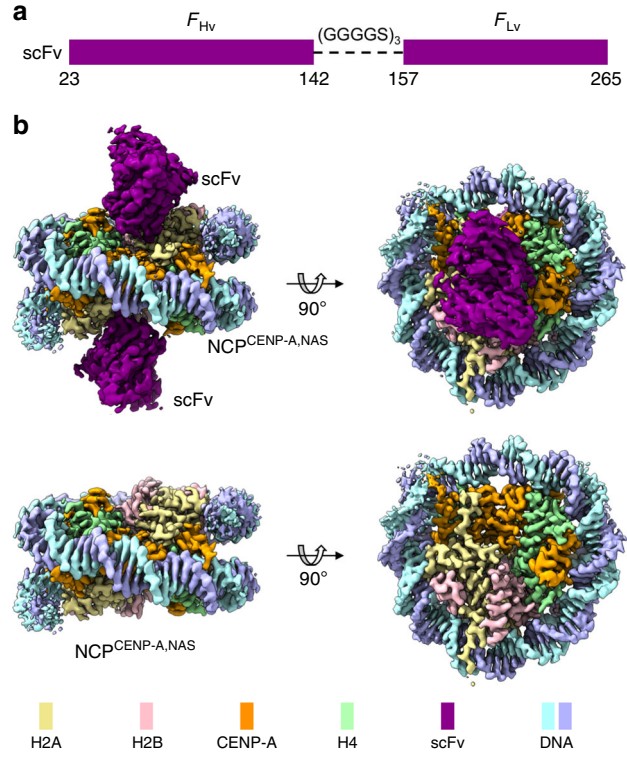

**Fig. 1** Cryo-EM density map of the native-like CENP-A nucleosome assembled with a native α-satellite DNA in complex with scFv. **a** Organization of the single chain antibody variable fragment (scFv): heavy chain ($F_{Hv}$) and light chain ($F_{Lv}$) are linked by three repeats of GGGGS. **b** Cryo-EM density map at 2.6 Å resolution: the scFv-$NCP^{CENP-A, NAS}$ complex (upper) and $NCP^{CENP-A, NAS}$ only (lower). The maps were generated in ChimeraX[50]

and 2a, c). scFv residue Arg124 serves as an "anchor" by insertion into the pocket of the acidic patch, forming salt bridges as well as hydrogen bonds with acidic patch residues Glu61, Asp90, and Glu92 of H2A (Fig. 2c). The arginine at this location is conserved and binds to the acidic patch similarly in previously studied nucleosome–protein complexes[3,11–16] (Supplementary Fig. 7). Additional electrostatic interactions are formed between scFv Arg126 and Glu113 of H2B and between scFv Arg188 and Glu64 of H2A. Unique to this complex, many scFv residues form hydrogen bonds with residues in H2A and H2B (Fig. 2a, c), including scFv Asn52 and Tyr76 with H2A Glu91 and Asn94, respectively; scFv Tyr74 with H2A Glu91, scFv Ser123 with H2A Asp90, scFv Tyr54 and Ser127 with H2B Glu105 and H109, respectively; and scFv Tyr190 with H2A Glu64. These tight and specific interactions might help stabilize the nucleosome structure during cryo-sample preparation by preventing the first step of nucleosome dissociation, involving H2A–H2B interactions with the DNA end regions of the nucleosome[17].

**Binding of the scFv does not perturb nucleosome structures.** Comparison of our cryo-EM $NCP^{CENP-A, NAS}$ structure with the CENP-A NCP crystal structure consisting of a palindromic one-half of the human α-satellite DNA (PAS)[18] revealed that the histone octamer conformations (in the defined regions of both structures) to be very similar, with a root mean square deviation of less than 0.6 Å (Supplementary Fig. 8); thus, scFv binding had little perturbation to the NCP structure. We further determined the cryo-EM structure of the free $NCP^{CENP-A, NAS}$ in the absence of the scFv (Supplementary Fig. 9 and Supplementary Table 1) as a control. The free CENP-A nucleosome showed substantial dissociation comparing to the scFv bound nucleosomes

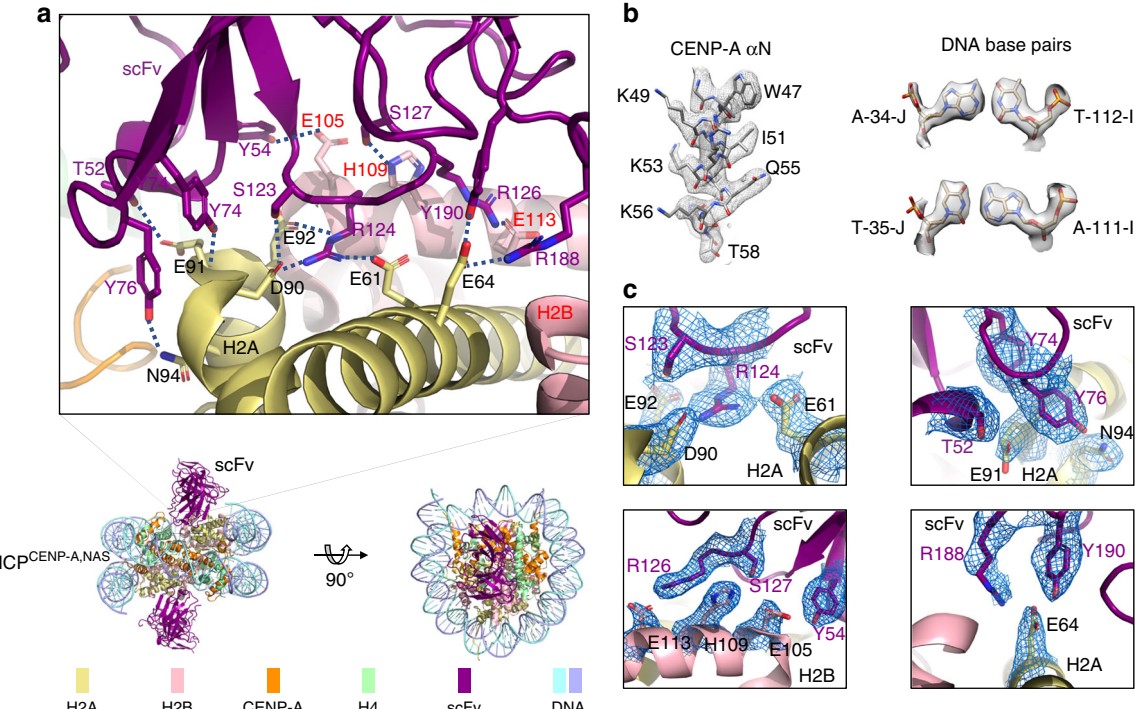

**Fig. 2** Overall structure of the scFv-NCP^CENP-A, NAS complex and interactions between scFv and NCP^CENP-A, NAS. **a** The overall structure of the scFv-NCP^CENP-A, NAS complex. Enlarged region shows the detailed interactions between scFv and the H2A–H2B dimer. Dashed lines show hydrogen bonds with distances less than 3.0 Å. Side chains are shown in sticks. Oxygen and nitrogen atoms are colored in red and blue, respectively. Residues in scFv, H2A, and H2B are labeled in magenta, black, and red, respectively. **b** Illustration of density maps for assignment of typical DNA base pairs, and a region in αN-helix of CENP-A. Maps were prepared in Chimera[47]. **c** Cryo-EM density maps of the scFv and H2A–H2B residues that form direct interactions as shown in (**a**), contoured at 3σ. Both (**a**) and (**c**) were prepared using PyMOL

(Supplementary Fig. 4d). We selected 303,864 number of refined particles and reconstructed a density map at the 3.40 Å resolution, which is insufficient to define the DNA bases unambiguously. We also found that the resolution of the density map only improves slightly (0.05 Å) when the particle numbers were increased from 220,908 to 303,864, suggesting that the resolution limiting factor is particle heterogeneity rather than numbers. Nevertheless, the free CENP-A nucleosome density map and structure fit the corresponding map and structure of the CENP-A nucleosome bound to the scFv very well (Supplementary Fig. 10), demonstrating again that binding of the scFv did not perturb the nucleosome structure.

**Structural features of the native-like CENP-A nucleosome.** In our NCP^CENP-A, NAS structure, the DNA regions at the entry and exit sites are well defined; residue Lys49 in the αN helix of the CENP-A and Arg42 of the proceeding region interact with the backbone phosphates of the DNA near the entry and exit sites, while CENP-A residue Arg43 inserts into a DNA minor groove and Arg44 interacts with the backbone phosphate near the dyad (Fig. 3a). By contrast, the 13 bp DNA at each corresponding end was missing in the crystal structure. The well-structured 145 bp DNA in our cryo-EM structure is supported by previous hydroxyl radical foot-printing results, which showed that the cleavage pattern for the entry/exit DNA is similar to that of the DNA region deep within the CENP-A NCP[19]. Moreover, 145 bp DNA was reported as the major fragment observed for CENP-A nucleosomes assembled on its native centromere sequence in an MNase digestion experiment[20]. Notably, the ordered DNA ends were also observed in the cryo-EM structures of the NCP containing CENP-A and 147 bp W601 DNA (NCP^CENP-A, W601) bound to kinetochore protein CENP-N, albeit with much lower

local resolutions[21–23], confirming again that the structured DNA ends are not induced by scFv binding.

We also solved the structure of the scFv-NCP^CENP-A, W601 complex at a resolution of 2.63 Å (Supplementary Fig. 11 and Supplementary Table 1). NCP^CENP-A, W601 and NCP^CENP-A, NAS showed similar structures at the DNA ends but displayed substantial conformational differences in super-helical location (SHL) 1 to 2 (Supplementary Fig. 12). In addition, we observed a large deviation of the DNA structure in SHLs 1.5 to 2.7 and −1.5 to −2.7 between the cryo-EM NCP^CENP-A, NAS and crystal CENP^CENP-A, PAS structures (Fig. 3b). It appears that the DNA conformation at the SHL 1 to 3 region is more variable than those at other locations. Intriguingly, this region is also the binding site of the ATPase domain of chromatin remodeling enzymes[24]. Moreover, the C-terminal tail of CENP-A with residues LEEGLG were well resolved in both NCP^CENP-A, W601 and NCP^CENP-A, NAS cryo-EM structures (Fig. 3c and Supplementary Fig. 13). The two Leu residues form similar hydrophobic interactions with each other compared to analogous Ile and Leu residues in the structure of NCP^H3-IEEGLG, W601 in complex with the CENP-C motif (Supplementary Fig. 13b)[12]. The two Leu residues are missing in the CENP-A crystal structure (Fig. 3c)[18]. It is likely that crystal packing and lower resolution of the crystal structure may contribute to these observed differences.

Alignment of the human NCP^CENP-A, NAS and NCP^H3.1, PAS structures[25] shows that the DNA ends in the NCP^CENP-A, NAS are shifted outwards by ~6 Å (Fig. 4a). The large side chain of Trp47 in the αN helix of CENP-A, which corresponds to H3.1 Ala47, packs against the side chain of H4 Lys44 and is likely responsible for a slightly outward shift of the αN helix of CENP-A (Fig. 4b). In addition, the αN helix is a half turn shorter in CENP-A than in H3.1, forcing the Arg42 residue in the preceding region of

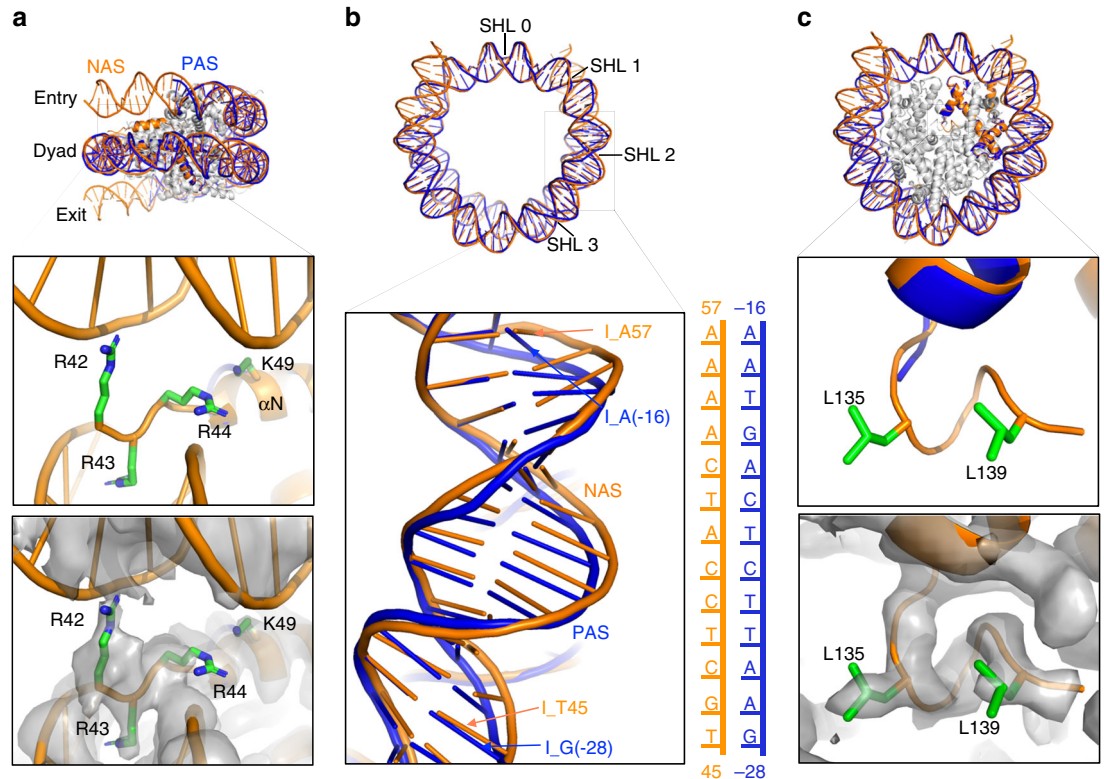

**Fig. 3** Comparison of the cryo-EM and crystal structures of the CENP-A NCPs. **a** Side view (top) of the structures of cryo-EM NCP^CENP-A, NAS (orange) and crystal NCP^CENP-A, PAS (PDB ID: 3AN2) (blue) aligned using their core histones. Enlarged region (middle) shows the DNA end region. Residues Arg42 and K49 of CENP-A interact with the entry/exit DNA while residue Arg43 of CENP-A inserts into the DNA minor grove and Arg44 interacts with the backbone of the DNA close to the dyad region in the NCP^CENP-A, NAS structure. The density map of the same region (bottom), contoured at 2.7σ. **b** Top view (top) of the cryo-EM NCP^CENP-A, NAS (orange) and crystal NCP^CENP-A, PAS (blue) DNA structures aligned using their core histones (not shown). Enlarged region shows the conformational difference of DNA in the two structures at the SHLs 1.7–2.5 region. **c** Top view (top) of the cryo-EM NCP^CENP-A, NAS (orange) and crystal NCP^CENP-A, PAS (blue) structures aligned using their core histones. Enlarged region (middle) shows the C-terminal tail (LEEGLG) region of CENP-A in the cryo-EM structure, which is missing in the X-ray structure. Density map of the same region (bottom), contoured at 3σ

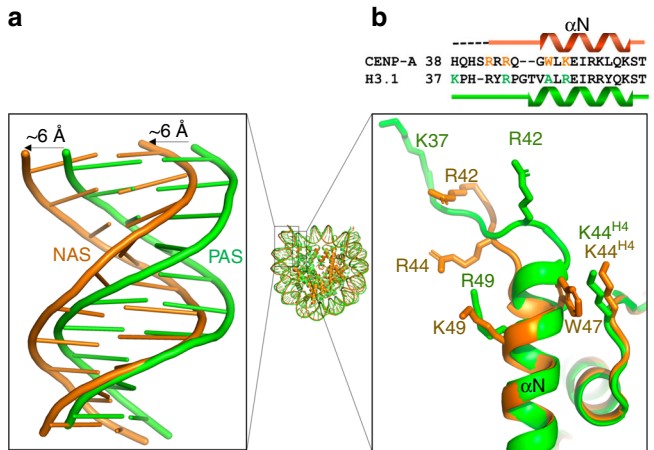

**Fig. 4** The CENP-A nucleosome has a more open DNA conformation. **a** The cryo-EM structure of NCP^CENP-A, NAS (orange) and the crystal structure of human NCP^H3.1, PAS (green, PDB ID: 5AV6) were aligned using the core histones. Enlarged region shows the comparison of the DNA end regions. **b** Sequence alignment of the human CENP-A and H3.1 αN region with secondary structure element (top). Close-up view of the αN helix regions of CENP-A (orange) and H3.1 (green) (bottom). Side-chains of charged residues in CENP-A or H3.1 that interact with nucleosome DNA ends are shown as sticks. Side-chain of CENP-A Trp47 corresponding to Ala47 in H3.1 packs against the side-chain of H4 Lys44

CENP-A to move outward to form interactions with the end DNA (Figs. 3a and 4a). The more closed DNA conformation in the NCP^H3.1, PAS structure is unlikely caused by crystal packing as the cryo-EM structure of H3 nucleosome displays the same conformation (Supplementary Fig. 14).

## Discussion

Our cryo-EM NCP^CENP-A, NAS structure has implications for the structure and function of centromeric chromatin. The outward shifts of the DNA at the entry and exit sites in the CENP-A NCP would lead to a more open conformation of the linker (or flanking) DNA in the nucleosome in comparison with those in the H3 nucleosome bound to a linker histone (Fig. 5a)[26,27]. Binding of the linker histone to the more open CENP-A nucleosome would require additional bending of the linker DNA, which is energetically unfavorable. Thus, our structure provides a possible explanation for the earlier observation that linker histones bind weakly to the CENP-A nucleosome and are largely absent at the centromeric chromatin[19]. Since linker histones help condense chromatin, the lack of linker histones would lead to a less condensed chromatin structure at the centromere, which in turn could make the CENP-A nucleosome more accessible for binding of kinetochore proteins such as CENP-C[12] and CENP-N[21,22] for kinetochore assembly. We note that the more open DNA end conformation of the CENP-A nucleosome is determined by the residues of the CENP-A histone (Fig. 4b)[18–20] instead of DNA sequence. Thus, the active centromeric nucleosome containing CENP-B box DNA sequence, which is recognized by CENP-B[28], is

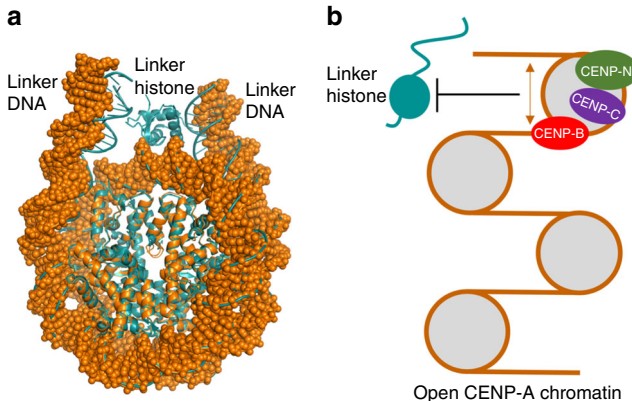

**Fig. 5** Implications of the cryo-EM NCP$^{CENP-A, NAS}$ structure for the structure and function of centromeric chromatin. **a** A model of the CENP-A nucleosome (orange) with 10 bp linker DNA at each end, which is built by aligning a 10 bp DNA of a 20 bp B-DNA to the 10 bp DNA at each end of the DNA in NCP$^{CENP-A, NAS}$. The CENP-A nucleosome was overlaid on the nucleosome bound to the globular domain of the linker histone H5 (cyan, PDB ID: 4QLC) by aligning the core histones. **b** Diagram illustration of the hypothesis that the active centromeric nucleosomes contain CENP-B box DNA sequence is likely to have more open DNA conformation (indicated by the arrow), which would prevent linker histones from binding to the CENP-A nucleosome stably and lead to less condensed chromatin structure that is more suitable for binding of kinetochore proteins CENP-C[12], CENP-N[21,22], and CENP-B[28]

likely to have an open structure as well. Consistent with this hypothesis, the lack of linker histones at centromere was previously shown to be required for mitotic fidelity[19] as kinetochore assembly at the centromere is critical for chromosome segregation. Notably, the lack of linker histones at centromere was previously attributed to the flexibility of DNA ends in the CENP-A nucleosome based on the crystal structure of the NCP$^{CENP-A, PAS}$ and the ~20 Å resolution cryo-EM 3D models of the CENP-A nucleosome[19].

In this work, we demonstrate that the scFv can be used to stabilize nucleosomes for determination of atomic resolution native-like nucleosome structures by cryo-EM. We have used this method to provide insights into the structure and function of the human CENP-A nucleosome with native DNA sequence. Determination of nucleosome structures by cryo-EM avoids the crystal packing at the DNA ends, which may alter the number of nucleotides associated with core histones[1–3]. Three factors may have contributed to the high resolution density map of the scFv–nucleosome complex: (i) scFv binding reduces the background noise of the cryo-EM micrograph by preventing nucleosome dissociation; (ii) the unique well-defined hydrogen bond network formed between the scFv and NCPs could make the structure of the complexes more rigid, allowing accurate alignment of the particles; (iii) the larger size of the scFv–nucleosome complex (relative to the free nucleosome) increases the signal/ noise ratio of the particle data.

The binding properties of the scFv to the nucleosome parallels the described interactions of mAb PL2-6 with the "epichromatin" region of interphase nuclei from diverse eukaryotic species[29]. As acidic patch residues are broadly conserved, we anticipate the scFv can be applied to the structural determination of nucleosomes from different species. For example, the scFv approach may be readily used to determine the structures of a number of nucleosomes that play important functional roles: the centromeric nucleosome of budding yeast containing the AT-rich DNA[30], the nucleosome containing the specific binding site of pioneer transcription factors[31,32], the nucleosome with DNA

sequences favored by intasomes for integration of virus DNA into human genomes[33,34]. It is expected that the scFv should also be helpful for structural determination of nucleosomes bound to other protein factors whose binding sites on the nucleosome do not overlap with that of the scFv, for example, the chromatosomes[26,27]. Finally, high-resolution structures of NCPs with differing native DNA sequences are ultimately required to understand how DNA sequences affect nucleosome positioning and dynamics in the genome.

## Methods

**scFv design, cloning, expression, purification, and refolding**. Design of scFv was based on a previous single chain antibody structure (PDB ID: 2GKI)[9]. The DNA sequence encoding the mouse mAb PL2-6 antibody heavy chain variable region (GenBank id: X60334) and light chain variable region (GenBank id: X60341) and a linker with three repeats of GGGGS were optimized for *Escherichia coli* usage, synthesized commercially (BioBasics, NY) and was subcloned into pET His$_6$-TEV vector from Addgene plasmid #48284. Expression of recombinant scFv was made using BL-21 *RIPL* CodonPlus cells. Protein expression was induced by adding 0.2 mM isopropyl β-d-1-thiogalactopyranoside (IPTG) for 18 h at 25 °C. Cells were then lysed by suspension in B-per Bacterial Protein Extraction Reagent (Thermo Fisher Scientific, IL). Inclusion bodies that contain scFv were harvested by centrifugation at 15,000$g$ for 30 min. Refolding of antibody fragments from inclusion bodies followed the procedures described previously[35]. Briefly, the inclusion bodies were denatured and reduced in GTE buffer (6 M Guanidine, 100 mM Tris–HCl, pH 8.0, 2 mM EDTA) with 10 mg/ml 1,4-dithioerythritol (Millipore Sigma, MO). Refolding was carried out by quickly mixing reduced/denatured antibody fragment solution with 100-fold buffer A (100 mM Tris–HCl, pH 9.5, 1 mM EDTA, 0.5 M arginine, 551 mg/L oxidized glutathione) at 10 °C for 48 h. The refolding solution was then dialyzed against 50-fold buffer B (20 mM Tris–HCl, pH 7.4) with 100 mM urea at 4 °C for an additional 24 h. The dialyzed sample was then filtrated through a 0.22-μM filter to remove aggregates and bound to 4 ml SP sepharose fast flow beads (GE Healthcare). The SP beads were transferred to an Econo column (Bio-Rad) and washed with 150 ml buffer B and 20 ml buffer B with 100 mM NaCl. scFv was eluted using 50 ml buffer B with 250 mM NaCl and further purified using metal-affinity chromatography with a 5-ml His-trap column (GE Healthcare) using AKTA FPLC system. The folded scFv was concentrated and purified using a Superdex S75 10/300 GL gel filtration column in buffer (20 mM Tris–HCl, pH 7.4, 1 mM EDTA and 150 mM NaCl) at 4 °C.

**Preparation of nucleosomal DNAs**. The 145 bp human α-satellite DNA was amplified by PCR from a previous plasmid (a gift from professor Ben E. Black). To increase the yield of the α-satellite DNA, we generated a modified pUC18 plasmid harboring 12 copies of 145 bp human α-satellite DNA sequence separated by EcoRV (GATATC) cleavage sites. The plasmid used to produce 147 bp Widom 601 DNA was a gift from professor Tan Song. Plasmids bearing human α-satellite DNA or Widom 601 DNA were produced in Alpha-Select Chemically Competent cell (Bioline). Harvested cells were treated with lysis buffer (1% SDS + 0.2 M NaOH) and neutralization buffer (4 M potassium acetate, 2 M acetic acid). after centrifugation at 8000$g$ for 30 min, the plasmids in supernatant were precipitated, redisolved, and further purified by phenol and chloroform extraction. 145 bp human α-satellite DNA or 147 bp Widom 601 DNA were cleaved from the plasmid by EcoRV (NEB) digestion[36].

The sequence of the 145 bp human α-satellite DNA is as follows:

ATCAATATCCACCTGCAGATTCTACCAAAAGTGTATTTGGAAACTGC TCCATCAAAAGGCATGTTCAGCTCTGTGAGTGAAACTCCATCATCACAA AGAATATTCTGAGAATGCTTCCGTTTGCCTTTTATATGAACTTCCTGAT

**Histones purification and nucleosomes reconstitution**. *Drosophila* or human core histones H2A, H2B, H3, H4 were produced as recombinant protein in *E. coli* BL21-CodonPlus(DE3)-RPIL (Novagen). Expressed histones were purified from the inclusion body using Hitrap SP column with AKTA FPLC (GE Healthcare) under denaturing condition[36]. Human CENP-A was expressed with an N-terminal his6 tag in Alpha-Select Chemically Competent cell (Bioline). Expressed CENP-A was purified by Ni affinity chromatography using Ni-NTA beads (QIAGEN) under denaturing condtion[18]. His6 tag were removed by thrombin protease (GE Healthcare) cleavage under non-denaturing condition (5 mM Tris–HCl, pH 7.4 and 5 mM 2-mercaptoethanol). All histones were further purified with a RP-protein column (Waters) and lyophilized. Nucleosomes were assembled by mixing histone octamer with DNA at 1:1.2 molar ratio in 10 mM Tris–HCl, pH 8.0, 1 mM EDTA (TE10/1) and 2 M NaCl, followed by dialysis in the same buffer with a gradual decrease in the NaCl concentration from 2 M to 0.25 M over 18 h. The CENP-A nucleosome assembled with human α-satellite DNA was heat re-positioned by incubating the reconstituted nucleosome at 37 °C for 3 h. All nucleosomes were purified from extra free DNA using DEAE-5PW column with AKTA FPLC (GE Healthcare)[36].

**Preparation of the scFv and nucleosome complex**. The ratio of scFv to nucleosome was empirically determined by titration of scFv with nucleosomes and visualized by native PAGE gel. scFv (~5 μM) was gradually added to nucleosome (~0.5 μM) in 10 mM Tris–HCl, pH 7.4 and 1 mM EDTA with 30 mM NaCl. The complex was concentrated to ~10 μM for cryo-EM analysis.

**Electrophoretic mobility shift assay**. Binding reactions of scFv and NCP were carried out on ice for 10 min in buffer containing 10 mM Tris, 1 mM EDTA, 30 mM NaCl, 1% Ficoll 6000 at pH 7.4. Reactions contained 100 nM NCP, and either 100, 200, or 300 nM scFv. 10 μl of the solution was analyzed on a 5% acrylamide gel in 0.2× TBE, and run at 100 V for 120 min at 4 °C. After electrophoresis, gels were stained with Midori Green Advance (Bulldog) and the gel images were visualized using ImageJ (http://imagej.nih.gov).

**Stability assay of CENP-A NCP with native DNA in complex with scFv**. 100 nM of NCP$^{CENP-A, NAS}$ in the absence of scFv or the presence of three folds of scFv was incubated in reaction solution (10 mM Tris pH 7.4, 1 mM EDTA, 1% Ficoll 6000) with 0, 50, 100, or 150 mM NaCl overnight at 4 °C. 10 μl of the each reaction was loaded on a 5% acrylamide gels in 0.2× TBE, and run at 100 V for 120 min at 4 °C. The gel was imaged as described above.

**ITC experiments**. ITC experiments were performed on a PEAQ-ITC micro-calorimeter (Malvern) at 25 °C. scFv and nucleosome samples were extensively dialyzed against the ITC buffer (20 mM Tris–HCl, pH 7.4, 50 mM NaCl, 1 mM EDTA) and degassed before loading into the syringe and the cell. In a typical titration experiment, 0.5 μM of the nucleosome was titrated with 11.3 μM scFv in the ITC buffer. The ITC data was analyzed using MicroCal PEAQ-ITC data analysis software (Malvern). Binding curves were generated by plotting the heat change of the binding reaction against the ratio of the total concentration of scFv to the total concentration of the nucleosomes. The association constant (Ka) and the stoichiometry of binding (n) were determined by fitting the observed binding curves to the model that two molecules scFv bind to the two sides of one nucleosome independently.

**Cryo-electron sample preparation and imaging**. Cryo-EM grid preparations were performed by applying 3 μl of freshly prepared NCP or scFv-NCP complexes at a concentration of 7.5 μM to a glow-discharged Quantifoil R1.2/1.3 holy carbon copper grids. The grids were vitrified by plunging into liquid ethane with a Vitrobot Mark IV (FEI) operated at 22 °C and 70% humidity. The frozen grids were stored in liquid nitrogen until data collection. Cryo-EM data collection were acquired on a FEI Titan Krios operated at 300 kV and equipped with a K2 Summit direct detector (Gatan). Movies were recorded at super-resolution mode at a dose rate of 2.63 e$^-$/Å$^2$/s with a total exposure time of 15.2 s, for an accumulated dose of 40 e$^-$/Å$^2$. Intermediate frames were recorded at every 0.4 s for a total number of 38 movie frames per micrograph. Defocus value ranged from −0.8 to −2.0 μm with a step size of 0.2 μm. The physical pixel sizes of the H3 and CENP-A NCPs were 1.72 and 1.358 Å, respectively.

**Image processing**. For scFv-NCP$^{CENP-A, NAS}$ dataset, movie frames were aligned and gain corrected using MotionCor2[37] with 5 by 5 patches and B-factor of 100. Whole micrograph CTF estimation was performed using GCTF[38] or CTFFIND4[39]. A subset of 10 micrographs was 10 Å lowpass filtered[40] for particle picking using crYOLO 1.0.4[41] with the general network model in 4 × 4 patch mode, the picked particles were manually checked and those in ice contaminated area were excluded using cryolo_boxmanager.py[41]. A model was trained with these 10 micrographs using crYOLO, and subsequently was used for picking on all 1416 micrographs. A total of 520,603 particles were picked and imported into RELION 3 beta2[42] and extracted using a box size of 320 pixels (resampled to box of 160 pixels for fast processing) and subjected to two initial rounds of 2D classification to identify and discard false positives or other apparent contaminants. Following 2D classification, 456,404 particles were imported into CryoSPARC v2 for Ab-initio 3D reconstruction[43], the resulting map was used as the reference for an initial 3D auto refine in RELION 3. Refined particles were re-extracted and re-centered, followed by another round of 2D classification, particles from classes showing secondary structure features were combined, and duplicate particles (distance below 100 Å) were removed. The resulting 435,083 refined particles were subject to one round of 3D classification with 8 classes. Particles of 4 classes that warrant high-resolution reconstruction were re-extracted using a larger box of 384 pixels (resampled to the box of 256 pixels for fast processing), followed by one round of 3D auto refinement. The new methods in RELION 3, Bayesian polishing and per-particle CTF refinements, were utilized for further processing, CTF refinements were iterated until the 3D refinement converged. The final map was sharpened using a B-factor of −24 Å$^2$. A flowchart of the data processing is shown in Supplementary Fig. 5. scFv-NCP$^{H3, W601}$, scFv-NCP$^{CENP-A, W601}$, and NCP$^{CENP-A, NAS}$ without scFv datasets were processed in the same way as described above for the scFv-NCP$^{CENP-A, NAS}$ dataset. The numbers of particles used in the final 3D refinement for each sample are listed in Supplementary Table 1. All resolutions were estimated by applying a soft mask around the protein density and the gold-standard FSC = 0.143 criterion, local resolutions of the maps were calculated using Resmap[44].

**Model building and structural analysis**. The atomic models of the W601 DNA (PDB ID: 3LZ0)[45], octameric histones (PDB ID: 1KX5)[46], octameric CENP-A histones (PDB ID: 3AN2)[18], scFv (PDB ID: 2GKI)[9] as rigid body were fitted into the corresponding 3D density maps using UCSF Chimera[47]. The coordinates including DNA, histone octamer, and scFv were further adjusted manually using COOT[48]. Most of the side-chain densities were clearly visualized throughout the map, which allowed unambiguous building and refining of the model. Mutations were also made according to the actual residues or DNA sequence used in this study. Structure models were further refined using Real Space Refine module in the Phenix suite[49]. EM maps and fitted structural models were deposited in EMDB and PDB. Figures were made using USCF Chimera[47], ChimeraX[50], or PyMOL 2.1.1. (The PyMOL Molecular Graphics System, Version 2.0 Schrödinger, LLC). Structure deviations were calculated using rms command in PyMOL. Electrostatic potential of nucleosome surface was calculated using APBS (http://www.poissonboltzmann.org/) plugin in PyMOL. Local R.M.S.D. was calculated using colorbyrmsd.py (https://pymolwiki.org/index.php/ColorByRMSD) in pyMOL.

**Reporting summary**. Further information on research design is available in the Nature Research Reporting Summary linked to this article.

## Data availability

Three-dimensional cryo-EM density maps have been deposited in the Electron Microscopy Data Bank under accession numbers EMDB-8938 (scFv-NCP$^{H3, W601}$), EMDB-8945 (scFv-NCP$^{CENP-A, W601}$), EMDB-8949 (scFv-NCP$^{CENP-A, NAS}$), and EMDB-0586 (NCP$^{CENP-A, NAS}$). The coordinates of atomic models have been deposited in the Protein Data Bank under accession numbers 6DZT (scFv-NCP$^{H3, W601}$), 6E0C (scFv-NCP$^{CENP-A, W601}$), 6E0P (scFv-NCP$^{CENP-A, NAS}$), and 6O1D (NCP$^{CENP-A, NAS}$). All other relevant data supporting the key findings of this study are available within the article and its Supplementary Information files or from the corresponding authors upon reasonable request. The source data underlying Supplementary Figs. 1a, b, 2c, 5a, c, 6c, 9c, and 11c are provided as a Source Data file. A reporting summary for this Article is available as a Supplementary Information file.

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

## Acknowledgements

The authors thank Drs. Kylie Walter, Jiansen Jiang, and Jemima Barrowman for critical reading of the manuscript and helpful discussions, Shipeng Li for α-satellite and W601 DNA preparation, Dr. Mitchell Ho for suggestions on scFv refolding, Tara Fox and Htet Khant for help with using the Krios microscope. This study utilized the computational resources of the High-Performance Computing Biowulf cluster at the NIH (http://hpc.nih.gov). Our work was supported by the intramural research program of Center of Cancer Research, National Cancer Institute, National Institutes of Health.

## Author contributions

Y.B., D.E.O., and A.L.O. conceived the antibody-nucleosome structure project. B.-R.Z. engineered scFv, prepared all NCP or scFv-NCP complexes, and performed biochemical/biophysical studies. Y.B. and P.Z. initiated the cryo-EM study. K.N.S.Y. conducted the cryo-EM experiments with P.Z., observed the antibody-nucleosome stabilization effect under cryo-EM, and worked together with B.-R.Z. to optimize the stabilization condition. K.N.S.Y. and B.-R.Z. built the initial structural models using cryo-EM maps reconstructed in RELION 2.0 under the guidance of P.Z. B.C. collected cryo-EM data set for the NCP[CENP-A,NAS] sample. B.-R.Z. performed final processing of all data sets in RELION 3.0 and corresponding structural refinement. M.B. provided help with the quality control programs for data collection. J.H. provided a sample of CENP-A NCP with α-satellite DNA. B.-R.Z., K.N.S.Y., Y.B., and P.Z. analyzed structure and wrote the paper with comments from all authors.

## Additional information

**Competing interests:** The authors declare no competing interests.

