## [Peer Review File · Nature Communications]

Reviewers' comments:

Reviewer #1 (Remarks to the Author):

CENP-A is the centromere-specific histone H3 variant, and is an essential epigenetic marker for centromere identity. It has been known that CENP-A forms the centromere-specific nucleosome with histones H2A, H2B, and H4. The octameric CENP-A nucleosome structure has been reported by X-ray crystallography and cryo-EM. In this paper, Zhang and Bai Labs use cryo-EM to provide new evidence that the CENP-A forms an octameric nucleosome on the natural centromeric satellite sequence. This work provides some information that will be helpful in understanding the mechanism by which CENP-A forms the centromeric chromatin conformation. The authors found that the histone-binding, single chain antibody greatly improves the resolution of the cryo-EM nucleosome structure. The authors successfully determined the CENP-A nucleosome structure at 2.6 angstrom resolution, which is even higher than the previous crystal structure of the CENP-A nucleosome. This may provide an important tool for future nucleosome study with a cryo-EM method.

Major comments:

1. A major highlight of this work is that the CENP-A nucleosome structure containing the native centromeric satellite sequence was determined. There are many centromeric satellite sequences, and the functional one usually contains the CENP-B box sequences. However, the authors' structure does not contain the CENP-B box sequence. This may imply that the CENP-A nucleosome structure reported here may be an inactive one, and not an active one. Therefore, the DNA end flexibility of the CENP-A nucleosome without the CENP-B box may not be appropriate for direct comparison with the CENP-A nucleosome structure containing the CENP-B box sequence. This should be discussed in the text. In this respect, the biological significance of this work may not be substantial. Thus, the description for "active centromeric nucleosome architecture" may be an overinterpretation, and should be removed or rigorously toned down.

2. The histone-binding, single chain antibody used in this study greatly improved the resolution of the cryo-EM nucleosome structure. The authors successfully determined the CENP-A nucleosome structure at 2.6 angstrom resolution, which is even higher than the previous crystal structure of the CENP-A nucleosome. This may provide an important new tool for future nucleosome study with a cryo-EM method. It would be very nice, if the authors re-write the manuscript as a novel, methodological paper for future structural studies of the nucleosome, and not as a centromere biology paper.

3. I am not sure how the authors determined the DNA orientation in the nucleosome containing a non-palindromic sequence. Is it possible to make 2D average images with the same DNA orientation? If it was possible, what were the criteria for determining the orientation?

4. Authors determined the H3 nucleosome containing the Widom 601 DNA at 3.3 angstrom resolution. They used the RELION2.1 for the H3 nucleosome, but used the RELION3.0beta for the CENP-A nucleosome. The CENP-A nucleosome achieved much higher resolution than the H3 nucleosome, and the per-particle CTF refinement on the RELION3 is obviously critical for the resolution improvement. I suggest that authors also run the per-particle CTF refinement for the H3 nucleosomes containing the Widom 601 DNA to see if the resolution improves.

Minor comment:

5. The colors of local resolution maps are different in each map (Extended Data Figure3d,e, Extended Data Figure7d,e, Extended Data Figure11d,e). Authors should use the same colors and the same ranges for each of them.

Reviewer #2 (Remarks to the Author):

Cryo-EM structure of a native-like human centromeric nucleosome aided by an antibody fragment

Zhou et al.

In this manuscript, Zhou and colleagues present a high-resolution (2.6Å) cryo-EM structure of a nucleosome containing the H3 variant CENP-A bound to its native sequence, alpha satellite DNA. The structure was solved with a single-chain antibody fragment (scFv) that binds to the acidic patch found in histones H2A and H2B. To the best of this reviewer's knowledge, this is the highest-resolution structure of a nucleosome obtained by cryo-EM. The authors also solved structures of a standard nucleosome (with H3) bound to the Widom 601 sequence, and the CENP-A-containing nucleosome bound to the 601 sequence, both bound to the scFv.

The authors make 3 major claims in this manuscript: (1) There is a need for high-resolution structures of different types of nucleosomes, in particular those containing native sequences (as opposed to the engineered 601 sequence, which we now know leads to nucleosomes that do not reflect natural ones); these structures are required to understand how differences among nucleosome types affect chromatin dynamics and all nucleosome-based processes; (2) The scFv will be an important tool for solving these structures; and (3) The DNAs at the entry/exit sites of the CENP-A-containing nucleosome are more open than those of an H3-containing nucleosome, which the authors propose explains why linker histones bind weakly to CENP-A-containing nucleosomes and are absent from centromeric chromatin.

I am in complete agreement with claim #1, and this manuscript makes an important contribution in this regard, with three nucleosome structures, including a very high-resolution of the very important CENP-A-containing nucleosome with its native cognate DNA.

I do, however, have problems with the other two claims.

(2) The idea that the scFv could be a general tool to facilitate the structural determination of nucleosomes at high resolution by cryo-EM is an interesting one, and of potential broad impact. However, there is not enough data in this manuscript to support it. Because binding of the scFv to the nucleosome could potentially affect its structure (e.g. by preventing conformational changes in the region it stabilizes), it is important to show that its addition is indeed critical to obtaining higher resolution structures. The micrographs presented in the Supplementary data do suggest that the particle integrity and distribution is much better in the presence of the scFv. Are the micrographs without the scFv truly representative of what is always seen with those samples? The ice looks to be of lower quality both in terms of contamination and possible variability in its thickness. Is that a result of not having the scFv? Were the grids with and without the scFv made the same day in the same manner? The most important missing piece of data is a cryo-EM reconstruction of the CENP-A-containing nucleosome with the NAS sequence in the absence of the scFv. The question is whether the same number of particles leads to a much lower resolution structure or not. I think that an increase in resolution is a more compelling argument for using scFv than simply having a larger fraction of intact particles, which can be solved by collecting more data.

(3) Although the more open nature of the DNA ends in the CENP-A/NAS nucleosome would be a nice explanation for some of the observed features of centromeric chromatin, the comparison with the crystal structure of the H3/PAS nucleosome (PDB: 5AV6) is not a fair one. The DNA ends in 5AV6 are involved in crystal packing, both through base stacking with another nucleosome on the same plane, and through close backbone interactions with a nucleosome in the neighboring plane. Therefore, one cannot rule out the possibility that the position of the DNA ends in 5AV6 is driven by crystal packing rather than by the differences between CENP-A and H3. If the authors want to make this biologically important point, they need to solve the structure of the H3/NAS nucleosome by cryo-EM (with the scFv) so they can compare it to their structure of the CENP-A/NAS nucleosome.

Response to Reviewers' comments:

We are most grateful to the reviewers for their valuable feedbacks. In this revised manuscript, we included new data to strengthen the main conclusions of our study and address all of the concerns raised by the reviewers.

Specifically, we have made the following key changes in the revised manuscript.

1. As reviewer #1 suggested, we have revised our manuscript to emphasize the scFv methodology instead of publishing it as a centromere biology paper. We changed the title to "Atomic resolution cryo-EM structures of native-like nucleosomes aided by an antibody fragment" accordingly. We also discussed the factors that may have contributed to the high-resolution of the scFv-nucleosome cryo-EM maps.
2. As recommended by reviewer #2, we now include a cryo-EM reconstruction of the CENP-A nucleosome with the NAS sequence in the absence of the scFv. When we used around the same number of particles, free CENP-A nucleosome leads to a much lower resolution (3.40 Å) than that of the CENP-A nucleosome bound to the scFv (2.59 Å). The new map also supports that scFv binding does not change the structure of the CENP-A nucleosome. These new results further strengthen our conclusion that the scFv could be used as a general tool to facilitate the structural determination of nucleosomes with natural DNA sequences at high resolution by cryo-EM.
3. As suggested by reviewer #1, we reprocessed the data set of the scFv-nucleosome with W601 DNA using RELION 3.0, the resolution of scFv-NCP^{H3, W601} and scFv-NCP^{CENP-A, W601} cryo-EM maps are improved. The new maps and structures have been deposited to EMDB and PDB, respectively.

Reviewer #1

CENP-A is the centromere-specific histone H3 variant, and is an essential epigenetic marker for centromere identity. It has been known that CENP-A forms the centromere-specific nucleosome with histones H2A, H2B, and H4. The octameric CENP-A nucleosome structure has been reported by X-ray crystallography and cryo-EM. In this paper, Zhang and Bai Labs use cryo-EM to provide new evidence that the CENP-A forms an octameric nucleosome on the natural centromeric satellite sequence. This work provides some information that will be helpful in understanding the mechanism by which CENP-A forms the centromeric chromatin conformation. The authors found that the histone-binding, single chain antibody greatly improves the resolution of the cryo-EM nucleosome structure. The authors successfully determined the CENP-A nucleosome structure at 2.6 angstrom resolution, which is even higher than the previous crystal structure of the CENP-A nucleosome. This may provide an important tool for future nucleosome study with a cryo-EM method.

Reply:

We thank the reviewer for the positive comments.

Major comments:

1. A major highlight of this work is that the CENP-A nucleosome structure containing the native centromeric satellite sequence was determined. There are many centromeric satellite sequences, and the functional one usually contains the CENP-B box sequences. However, the authors' structure does not contain the CENP-B box sequence. This may imply that the CENP-A nucleosome structure reported here may be an inactive one, and not an active one. Therefore, the DNA end flexibility of the CENP-A nucleosome without the CENP-B box may not be appropriate for direct comparison with the CENP-A nucleosome structure containing the CENP-B box sequence. This should be discussed in the text. In

this respect, the biological significance of this work may not be substantial. Thus, the description for “active centromeric nucleosome architecture” may be an overinterpretation, and should be removed or rigorously toned down.

Reply:

As recommended, we have now rigorously toned down the description for “active centromeric nucleosome architecture” and discussed the possible conformation of DNA in the active centromeric nucleosome that contains CENP-B box DNA (please see page 8 - 9 in the revised manuscript and the figure legend of Fig. 5). We noted that the open structural feature of our CENP-A nucleosome is determined by the CENP-A protein, independent of DNA sequence. For example, the open conformation was also observed in the CENP-A nucleosome containing the W601 DNA sequence (CENP-A nucleosome with W601 DNA structure in our study and references 18-20) Therefore, the CENP-A nucleosome consisting of other alpha satellite DNA sequences such as the CENP-B box DNA will likely have an open structure as well.

2. The histone-binding, single chain antibody used in this study greatly improved the resolution of the cryo-EM nucleosome structure. The authors successfully determined the CENP-A nucleosome structure at 2.6 angstrom resolution, which is even higher than the previous crystal structure of the CENP-A nucleosome. This may provide an important new tool for future nucleosome study with a cryo-EM method. It would be very nice, if the authors re-write the manuscript as a novel, methodological paper for future structural studies of the nucleosome, and not as a centromere biology paper.

Reply:

We are thankful to the reviewer for the suggestion. We have revised the manuscript to emphasize the scFv methodology. Specifically, we changed the title to “Atomic resolution cryo-EM structures of native-like nucleosomes aided by an antibody fragment”. We added paragraph in the introduction (page 3) give brief summary of the conclusion. We collected a new cryo-EM data set of the CENP-A nucleosome without scFv. Using around the same number of particles, the free CENP-A nucleosome leads to a much lower resolution of 3.4 Å (please see Supplementary Fig. 9, 10; Supplementary Table 1 and page 5-6), which strengthen the main conclusion of our study that the scFv could be “*an important new tool for future nucleosome study with a cryo-EM method*”, as noted by the reviewer. We also discussed the factors that may have contributed to the high-resolution of the scFv-nucleosome cryo-EM maps (please see page 9).

3. I am not sure how the authors determined the DNA orientation in the nucleosome containing a non-palindromic sequence. Is it possible to make 2D average images with the same DNA orientation? If it was possible, what were the criteria for determining the orientation?

Reply:

Some regions of DNA in the scFv-NCP^{CENP-A, NAS} or scFv-NCP^{CENP-A, W601} map can reach 2.0 Å resolution (Supplementary Fig. 6, 11), allowing unambiguous model building. The DNA orientation was determined by testing both orientations. The non-palindromic sequence can only fit well in one orientation in such high-resolution regions. For example, DT-35 in chain J pairs with DA-111 in chain I in one orientation fitted the density map very well. However, when the orientation was reversed, DA-35 in chain I paired with DT-111 in chain J, which would not fit the same map density well (see figure below).

4. Authors determined the H3 nucleosome containing the Widom 601 DNA at 3.3 angstrom resolution. They used the RELION2.1 for the H3 nucleosome, but used the RELION3.0beta for the CENP-A nucleosome. The CENP-A nucleosome achieved much higher resolution than the H3 nucleosome, and the per-particle CTF refinement on the RELION3 is obviously critical for the resolution improvement. I suggest that authors also run the per-particle CTF refinement for the H3 nucleosomes containing the Widom 601 DNA to see if the resolution improves.

Reply:

As suggested, we have now reprocessed the data set with W601 DNA using RELION3 and the same pipeline as shown in Supplementary Fig. 5. Indeed, the overall resolutions of scFv-NCP^{H3,W601} and scFv-NCP^{CENP-A,W601} maps have been improved to 2.99 and 2.63 Å, respectively. The reprocessed maps and structures have been deposited to the EMDB and PDB (please see Supplementary Fig. 2, 11 and Supplementary Table 1).

Minor comment:

5. The colors of local resolution maps are different in each map (Extended Data Figure3d,e, Extended Data Figure7d,e, Extended Data Figure11d,e). Authors should use the same colors and the same ranges for each of them.

Reply:

We are thankful to the reviewer for pointing out this discrepancy. We have used the same colors and ranges for each of the density map in the revised manuscript (please see Supplementary Fig. 2, 6, 9, 11).

Reviewer #2 (Remarks to the Author):

Cryo-EM structure of a native-like human centromeric nucleosome aided by an antibody fragment

Zhou et al.

In this manuscript, Zhou and colleagues present a high-resolution (2.6Å) cryo-EM structure of a nucleosome containing the H3 variant CENP-A bound to its native sequence, alpha satellite DNA. The structure was solved with a single-chain antibody fragment (scFv) that binds to the acidic patch found in histones H2A and H2B. To the best of this reviewer's knowledge, this is the highest-resolution structure of a nucleosome obtained by cryo-EM. The authors also solved structures of a standard nucleosome (with H3) bound to the Widom 601 sequence, and the CENP-A-containing nucleosome bound to the 601 sequence, both bound to the scFv.

The authors make 3 major claims in this manuscript: (1) There is a need for high-resolution structures

of different types of nucleosomes, in particular those containing native sequences (as opposed to the engineered 601 sequence, which we now know leads to nucleosomes that do not reflect natural ones); these structures are required to understand how differences among nucleosome types affect chromatin dynamics and all nucleosome-based processes; (2) The scFv will be an important tool for solving these structures; and (3) The DNAs at the entry/exit sites of the CENP-A-containing nucleosome are more open than those of an H3-containing nucleosome, which the authors propose explains why linker histones bind weakly to CENP-A-containing nucleosomes and are absent from centromeric chromatin.

I am in complete agreement with claim #1, and this manuscript makes an important contribution in this regard, with three nucleosome structures, including a very high-resolution of the very important CENP-A-containing nucleosome with its native cognate DNA.

Reply:

We appreciate the positive comments from the reviewer.

I do, however, have problems with the other two claims.

(2) The idea that the scFv could be a general tool to facilitate the structural determination of nucleosomes at high resolution by cryo-EM is an interesting one, and of potential broad impact. However, there is not enough data in this manuscript to support it. Because binding of the scFv to the nucleosome could potentially affect its structure (e.g. by preventing conformational changes in the region it stabilizes), it is important to show that its addition is indeed critical to obtaining higher resolution structures. The micrographs presented in the Supplementary data do suggest that the particle integrity and distribution is much better in the presence of the scFv. Are the micrographs without the scFv truly representative of what is always seen with those samples? The ice looks to be of lower quality both in terms of contamination and possible variability in its thickness. Is that a result of not having the scFv? Were the grids with and without the scFv made the same day in the same manner? The most important missing piece of data is a cryo-EM reconstruction of the CENP-A-containing nucleosome with the NAS sequence in the absence of the scFv. The question is whether the same number of particles leads to a much lower resolution structure or not. I think that an increase in resolution is a more compelling argument for using scFv than simply having a larger fraction of intact particles, which can be solved by collecting more data.

Reply:

We have observed that in general the quality of the micrographs without the scFv always looks worse than that with the scFv under the exact same freezing conditions. Free nucleosomes always show substantial dissociation (please see updated Supplementary Fig. 4d for two typical micrographs).

As recommended, we have now determined the cryo-EM structure of the CENP-A nucleosome with NAS in the absence of the scFv, using the exact same condition (freezing, type of grid, microscope and camera, imaging parameters) as that for the CENP-A nucleosome with the scFv. When we did reconstruction using around the same number of particles (303,864 without scFv compare to 301,644 with scFv), the free CENP-A nucleosome leads to a much lower 3.40 Å resolution (with scFv, 2.59 Å), insufficient to define the DNA bases unambiguously. Besides, we found that the density map resolution improved by only 0.05 Å when particles were increased from 220,908 to 303,864, indicating that collecting more data is not going to be very helpful to improve resolution.

We also found that at the given resolution, the structures of the CENP-A nucleosomes with and without scFv are nearly identical (protein backbone rmsd = 0.3 Å, DNA backbone rmsd = 0.8 Å, please see Supplementary Fig. 10), indicating that scFv binding does not perturb the nucleosome structure. We have included these new results in the revised manuscript (please see Supplementary Fig. 9,

Supplementary Table 1 and page 5-6). The corresponding map and structure have been deposited to the EMDB and PDB, respectively.

(3) Although the more open nature of the DNA ends in the CENP-A/NAS nucleosome would be a nice explanation for some of the observed features of centromeric chromatin, the comparison with the crystal structure of the H3/PAS nucleosome (PDB: 5AV6) is not a fair one. The DNA ends in 5AV6 are involved in crystal packing, both through base stacking with another nucleosome on the same plane, and through close backbone interactions with a nucleosome in the neighboring plane. Therefore, one cannot rule out the possibility that the position of the DNA ends in 5AV6 is driven by crystal packing rather than by the differences between CENP-A and H3. If the authors want to make this biologically important point, they need to solve the structure of the H3/NAS nucleosome by cryo-EM (with the scFv) so they can compare it to their structure of the CENP-A/NAS nucleosome.

Reply:

We thank the reviewer for pointing out the crystal packing issue of the H3/PAS nucleosome. We found that Cryo-EM structure of the H3 nucleosome in this study (with W601 DNA) and several other cryo-EM structures/maps of H3 nucleosomes (for example, 3.9 Å structure of the nucleosome core particle with W601 DNA (EMD-8140), 4.0 Å nucleosome with ALB1 enhancer DNA (EMD-6838)), all have closed DNA conformation. These results are consistent with the H3 nucleosome structures determined by X-ray crystallography, suggesting the closed DNA conformation in H3 nucleosomes is not due to crystal packing, as cryo-EM structures do not have crystal packing issue. We have now revised the manuscript to emphasize the scFv methodology instead of centromere biology as reviewer #1 suggested.

REVIEWERS' COMMENTS:

Reviewer #1 (Remarks to the Author):

Zhou and colleagues have addressed my concerns. In the revised manuscript, they have changed the title to emphasize the scFv methodology and added paragraphs for it. They have also improved the resolution of H3 nucleosomes using the Relion3. Now, I recommend this revised version for publication in Nature Communications.

Reviewer #2 (Remarks to the Author):

Here are my comments regarding how the authors addressed my concerns (2) and (3) in my original review.

(2) scFv as a tool to facilitate high-resolution structure determination.

I am satisfied that the additional data the authors provide make a strong case for the scFv to have significantly improved the resolution of their reconstruction.

I also agree with their reframing of their manuscript to emphasize the scFv methodology.

(3) The open nature of the DNA ends in the CNEP-A/NAS nucleosome.

I still find it inappropriate to use the crystal structure of human NCPH3.1,PAS (PDB: 5AV6) for the comparison when the authors know that DNA ends in that structure are not free. Furthermore, although the authors claim (p8, lines 168-171) that "... structure is unlikely caused by crystal packing as the cryo-EM structure of H3 nucleosome displays the same conformation (Supplementary Fig.3)." there is no data in Supplementary Fig.3 that makes this point. The figure shows a structure but does not compare it with others. To make this point, the authors need to show a direct comparison between 5AV6 and the structure shown in Supp.Fig.3, with one of the structures colored according to the local RMSD between them.

Response to Reviewers' comments:

Reviewer #2 (Remarks to the Author):

Here are my comments regarding how the authors addressed my concerns (2) and (3) in my original review.

(2) scFv as a tool to facilitate high-resolution structure determination.

I am satisfied that the additional data the authors provide make a strong case for the scFv to have significantly improved the resolution of their reconstruction.

I also agree with their reframing of their manuscript to emphasize the scFv methodology.

Reply:

We thank the reviewer for the positive comments.

(3) The open nature of the DNA ends in the CNEP-A/NAS nucleosome.

I still find it inappropriate to use the crystal structure of human NCPH3.1,PAS (PDB: 5AV6) for the comparison when the authors know that DNA ends in that structure are not free. Furthermore, although the authors claim (p8, lines 168-171) that "... structure is unlikely caused by crystal packing as the cryo-EM structure of H3 nucleosome displays the same conformation (Supplementary Fig.3)." there is no data in Supplementary Fig.3 that makes this point. The figure shows a structure but does not compare it with others. To make this point, the authors need to show a direct comparison between 5AV6 and the structure shown in Supp.Fig.3, with one of the structures colored according to the local RMSD between them.

Reply:

We thank the reviewer for the suggestions. We have now included a new supplementary figure (Supplementary Fig. 14) to show a direct comparison between crystal structure 5AV6 and cryo-EM structure of scFv-NCP^{H3, W601} shown in Supp.Fig.3. The two structures were aligned based on histone protein backbones, the backbones of the DNA ends in the two structures show very similar conformation. We also find that the DNA in 5AV6 structure fits well into the DNA density of our scFv-NCP^{H3, W601} cryo-EM map (Supplementary Fig. 14b), suggesting that despite of crystal packing, the DNA ends in 5AV6 structure displays the closed conformation.